# Influence of LDL-Cholesterol Lowering on Coronary Plaque Progression of Non-Target Lesions in Patients Undergoing Percutaneous Coronary Intervention: Findings from a Retrospective Study

**DOI:** 10.3390/jcm12030785

**Published:** 2023-01-18

**Authors:** Weiwei Quan, Hui Han, Lili Liu, Yi Sun, Zhengbin Zhu, Run Du, Tianqi Zhu, Ruiyan Zhang

**Affiliations:** 1Department of Cardiovascular Medicine, Ruijin Hospital, Shanghai Jiao Tong University School of Medicine, Shanghai 200025, China; 2Department of Cardiovascular Medicine, Shanghai Ninth People’s Hospital, Shanghai Jiao Tong University School of Medicine, Shanghai 200011, China

**Keywords:** LDL-C, plaque progression, non-target lesion, quantitative coronary angiography

## Abstract

The progression of NTLs after PCI accounts for a significant portion of future adverse cardiac events. The reduction in LDL-C reduces cardiovascular events. This has, however, not yet been shown in a real-world setting. We aimed to investigate the association between LDL-C changes with progression in NTLs. A total of 847 patients with successful PCI were enrolled. Patients with follow-up LDL-C ≥ 1.4 mmol/L or percent reduction <50% compared to baseline were Non-optimal group (*n* = 793); patients with follow-up LDL-C < 1.4 mmol/L and percent reduction ≥50% compared to baseline were Optimal group (*n* = 54). Compared to Non-optimal group, Optimal group presented a lower rate of NTL plaque progression (11.11% vs. 23.96%; *p* = 0.007) and a lower follow-up TC (2.77 ± 0.59 vs. 3.66 ± 0.97; *p* < 0.001) and LDL-C (1.09 ± 0.26 vs. 2.03 ± 0.71; *p* < 0.001). The univariate logistic regression analysis revealed that follow-up LDL-C < 1.4 mmol/L and a percent reduction ≥50% from baseline was a protective factor for NTL plaque progression (OR: 0.397; 95%CI: 0.167–0.941; *p* = 0.036). The multivariate logistic regression model revealed that follow-up LDL-C < 1.4 mmol/L and percent reduction ≥50% was indeed an independent factor associated with a lower rate of plaque progression of NTLs (OR: 0.398; 95% CI: 0.167–0.945; *p* = 0.037). Therefore, achieving guideline-recommended LDL-C level was associated with a significantly reduced risk of NTL plaque progression.

## 1. Introduction

Percutaneous coronary intervention (PCI) using drug-eluting stents has been demonstrated to remarkably reduce rates of re-stenosis and target lesion revascularization inside stented segments [1,2,3]. Patients with coronary artery disease (CAD) undergoing PCI are still at high risk of future adverse ischemic events, despite contemporary therapies. Previous studies reported that nearly half of the recurrent cardiac ischemic events were due to the progression of non-target lesions (NTLs) after successful PCI [4,5]. Secondary prevention of CAD after PCI is focused on optimal medical treatment and a strict control of cardiovascular risk factors.

Evidence from genetic, epidemiologic, and clinical studies have strongly demonstrated that low-density lipoprotein cholesterol (LDL-C) plays an essential role in the plaque progression in arteriosclerotic cardiovascular disease (ASCVD) patients [6]. Intensive lowering of LDL-C levels was found to be associated with improved event-free survival and to halt the atherosclerotic plaque progression observed by serial coronary angiography or intravascular ultra-sound (IVUS), or even drive the atherosclerotic regression [7,8,9]. However, studies assessing this relationship in trials of patients were often not representative of real-world clinical practice. Hence, whether the optimal control of LDL-C will halt the plaque progression of NTLs has not been well evidenced in the long-term treatment for secondary prevention after PCI from real-life patient cohort.

## 2. Materials and Methods

### 2.1. Study Population

This was a muti-center observational research study using data obtained from the RIPPER study (NCT 02692014). Data were collected in 19 coronary intervention centers of tertiary hospitals from 2541 patients who had completed at least two coronary angiographies (CAGs) in the same center within 12–24 months from 1 June 2013 to 1 December 2016.

The exclusion criteria were as follows: (1) inadequate clinical or laboratory data, or history of coronary artery bypass graft surgery (*n* = 370); (2) poor CAG image quality that does not meet the requirements of QCA (*n* = 380); (3) patients without any NTLs after single- or multi-vessel intervention (*n* = 944). In total, 847 patients were enrolled in this study (Figure 1A).

The study was carried out in accordance with the code of Ethics of the World Medical Association (Declaration of Helsinki). Written informed consent was obtained from all patients prior to their inclusion. The relevant protocols were approved by the Ethic Committees of all enrolling centers. Pharmaceutical and lifestyle changes were managed after PCI according to AHA/ACC guidelines [10,11].

### 2.2. Quantitative Coronary Angiography Analysis

Coronary angiograms were recorded at baseline and at follow-up periods. All datasets were transferred to the core laboratory in Ruijin Hospital affiliated to Shanghai Jiao Tong University School of Medicine. Coronary segments were defined according to the modified AHA/ACC coronary segment classification [12]. Quantitative coronary angiography (QCA) analysis was conducted by specialized technicians who were blinded to the identity and clinical characteristics of the patients according to a previous study.

All three major untreated coronary vessels including all side branches with a reference vessel diameter (RVD) of >1.5 mm in diameter were assessed by MEDCON TCS QCA software with similar angiographic views of NTL segments matched between baseline (BL) and follow-up (FU). Measured variables included the RVD, the minimal luminal diameter (MLD), and the diameter stenosis (DS). Delta DS was defined as the last DS minus the initial DS.

An NTL was defined as a stenosis lesion that was not responsible for the ischemic symptoms or positive functional ischemic test outcomes. For the purpose of this study, NTLs were identified as previously untreated lesions of 5-mm segment with percent diameter stenosis <50% on QCA.

Coronary plaque progression was defined when one of the following criteria was met: (1) ≥10% reduction in the diameter of a pre-existing lesion with ≥50% stenosis; (2) increase of ≥30% reduction in the diameter of a pre-existing lesion with <50% stenosis; (3) the development of a new stenosis with a ≥30% reduction in the diameter of a segment that was normal on the first angiogram, or the progression of any lesion to total occlusion on follow-up CAG [13]. A representative QCA result of NTL plaque progression was shown in Figure 1B.

### 2.3. Measurement of Clinical and Confounding Variables

Clinical data, including medical history of hypertension, diabetes mellitus, acute coronary syndrome, smoking status, body weight index (BMI), were attained at baseline. Blood pressure (BP), medications, total cholesterol (TC), triglycerides (TG), high-density lipoprotein cholesterol (HDL-C), low-density lipoprotein cholesterol LDL-C, fasting plasma glucose (FPG), glycated hemoglobin (HbA1C), c-reaction protein (CRP) were recorded at baseline and at follow-up. Diabetes was defined as fasting glucose ≥7.0 mmol/L, or treatment with oral hypoglycemic agents or insulin. Hypertension was defined as systolic blood pressure ≥140 mmHg and/or diastolic blood pressure ≥90 mmHg, or treatment with antihypertensive agents.

### 2.4. Statistical Analysis

The SPSS 22.0 (IBM Corporation, Armonk, NY, USA) was used for all statistics analyses. Continuous variables were compared using Student’s *t* tests or the Mann–Whitney U test as appropriate and are expressed as mean ± standard deviation; the categorical variables, compared with chi-square tests, are expressed as percentages. Serial changes were assessed using mixed modeling methodology adjusting for their baseline confounders. The significant variables (*p* < 0.05) were included in the logistic regression analysis to identify factors that were independently associated with NTL plaque progression. All *p* values were two-sided, and a *p* < 0.05 was considered statistically significant.

## 3. Results

### 3.1. Baseline Characteristics of the Study Subjects

A total of 847 patients were enrolled and classified into two groups based on achieved LDL-C levels at follow up. Patients with follow-up LDL-C ≥ 1.4 mmol/L or percent reduction less than 50% compared to baseline were Non-optimal group (*n* = 793); patients with follow-up LDL-C < 1.4 mmol/L and percent reduction ≥50% compared to baseline were Optimal group (*n* = 54).

The mean age of the study population was 63.1 ± 10.7 years; about 75.4% of the patients were men; 67.3% of the patients had hypertension; 59.3% had diabetes mellitus; 17.5% had history of ACS. The mean levels of baseline lipid parameters were TC 4.22 ± 1.21 mmol/L, LDL-C 2.46 ± 0.93 mmol/l, TG 1.85 ± 1.34 mmol/l, and HDL 1.03 ± 0.27 mmol/l. At discharge, about 97% of the patients were prescribed statins, 94.2% DAPT, 71.3% β receptor antagonist and 58.68% RASI/ARB therapy.

The two groups were similar with respect to age, clinical risk factors, biochemical results and treatments, except that the Optimal group had a less proportion of CKD (*p* < 0.001) and higher baseline LDL-C level compared to Non-optimal group (2.44 ± 0.91 vs. 2.84 ± 0.94; *p* = 0.004). Relative data were summarized in Table 1. Subgroup analysis was summarized in Appendix A.

### 3.2. Follow-Up Laboratory Test and QCA Measurements of NTLs

During average 14.7 ± 3.9 months (range, 12 to 24 months) follow up, coronary plaque progression occurred in 196 of the 847 patients (23.14%). Compared to the Non-optimal group, Optimal group presented a lower rate of plaque progression of NTLs (11.11% vs. 23.96%, *p* = 0.007) and a lower follow-up TC level (2.77 ± 0.59 vs. 3.66 ± 0.97; *p* < 0.001) and LDL-C level (1.09 ± 0.26 vs. 2.03 ± 0.71; *p* < 0.001). Subgroup analysis revealed that either of the single optimal subgroups (one with LDL-C < 1.4 mmol/L only, one with LDL-C reduction ≥50% only) did not present a lower rate of plaque progression of NTLs.

Other parameters like twice CAG procedure interval, fasting plasma glucose (FPG), HBA1C, triglyceride (TG), high density lipoprotein-cholesterol (HDL-C), serum creatinine (SCr) and uremic acid (UA) levels showed no difference between the two groups at follow-up periods. There was also no difference in lesion length, MLD, RVD, DS%, MLD DS% and Delta DS for NTLs between the two groups. Relative data were summarized in Table 1 and Table 2. Subgroup analysis was summarized in Appendix A.

### 3.3. Univariate and Multivariate Logistic Regression Analysis for the Presence of Plaque Progression of NTL

To establish whether follow-up LDL-C < 1.4 mmol/L and percent reduction ≥50% compared to baseline was an independent value in predicting plaque progression, we performed univariate (entry) and multivariate (forward) logistic regression analyses. Relevant data were summarized in Table 3.

According to the univariate logistic regression analysis, follow-up LDL-C < 1.4 mmol/L and percent reduction ≥50% compared to baseline was a protective factor for plaque progression of NTLs (OR: 0.397; 95%CI: 0.167–0.941; *p* = 0.036). After adjusting for traditional cardiovascular risk factors (including age, gender, smoking; history of hypertension, dyslipidemia, diabetes mellitus, chronic kidney disease and ACS) and follow-up clinical parameters (including twice CAG interval, FPG, HbA1C, TG, TC, HDL-C, LDL-C, SCr and UA), the multivariate logistic regression model revealed that follow-up LDL-C < 1.4 mmol/L and percent reduction ≥50% was indeed an independent factor associated with a lower rate of plaque progression of NTLs (OR: 0.398; 95% CI: 0.167–0.945; *p* = 0.037), suggesting that subjects with intensive LDL-C lowering strategy perceived lesser risk of plaque progression of NTLs.

## 4. Discussion

The main findings of the current analysis were as follows: 1) the risk of progression in NTLs still remained high with 23.14% in patients after PCI, although most of the patients had accepted optimal secondary prevention; 2) achieving LDL-C < 1.4 mmol/L and a percent reduction ≥ 50% was associated with a reduced risk of NTL angiographic progression; 3) after 12–24 months follow-up, only 54/847 (6.38%) patients had met the target goal of LDL-C control in this real-world RIFFER study in post-PCI patients from China.

Over the past decades, PCI with stent implantation has been a major therapeutic strategy for patients with acute and chronic coronary syndrome. Widespread use of drug-eluting stents and advances in interventional approaches, along with pharmacological therapies, efficiently improve paroxysmal and persistent myocardial ischemia, resulting in relief of symptoms, prevention of adverse cardiovascular events and better survival rates following PCI procedures [14,15]. Nonetheless, recurrent ischemic events are not wiped out. Particularly, recurrent major adverse cardiovascular events (MACEs) after acute coronary syndrome were attributed to non-culprit lesions in 11.6% of patients in the PROSPECT trial [4], equally attributed to culprit lesions. MACE outcomes are associated not only with severe coronary obstructions, but also with the vulnerability of plaque along the entire coronary tree [16]. It is of great value to identify the plaque anatomic characteristics of progressing non-culprit lesions to indicate the propensity of individual plaques to provoke a MACE. Regrettably, our database did not contain the morphological changes in plaques by IVUS or OCT due to technical reasons.

Although NTLs responsible for recurrent cardiac ischemic events are frequently characterized as mild, several studies have indicated that their stenosis remains the strongest predictor of future MACEs. This paradox might be attributed to severe plaque progression over time. In our study, QCA was used to detect atherosclerotic progression by assessing the narrowing of the coronary lumen. NTL progression was observed in 196 out of 847 patients by QCA. Notably, the incidence rate of plaque progression in NTLs was much lower in patients with optimal follow-up LDL-C level than in those without (11.11% vs. 23.96%; *p* = 0.007). Plaque progression, as suggested by previous studies, is a necessary step between early atherosclerosis and the cardiovascular event [17]. How to limit NTL plaque progression has become a major clinical issue with important therapeutic implications.

While primary and elective PCI procedures are proven to achieve the revascularization of the focal atherosclerotic lesions safely and efficiently, the systemic pro-atherogenic cardiovascular risk factors have been receiving increasing attention for the progressive disease. Plenty of evidence has demonstrated that LDL-C induces a causal and cumulative effect on the risk of CAD, though other lipids are also involved in the pathobiology of atherosclerosis [6]. Large-scale randomized controlled clinical trials explicitly suggested a significant reduction in CAD events with lipid-lowering medications [18,19,20]. In a real-world observational research study using data obtained from SWEDEHEART registry, patients with >50% LDL-C reduction in the early stage with 6–8 week post myocardial infarction (MI) would have a lower incidence of all outcomes compared with those without [21]. This result further supports previous randomized controlled clinical trial data, suggesting that great and earlier lowering of LDL-C after an MI confers the greatest benefit. Therefore, LDL-C is currently recommended as the primary lipid measurement for risk assessment and treatment guidance. Intensive LDL-C lowering therapies, including statins, proprotein convertase subtilisin/kexin type 9 (PCSK9) inhibitors and other lipid-lowering agents, might be a panacea to slow down NTL plaque progression after successful revascularization of target lesions by stent implantation. There is also evidence that lipid-lowering therapy tends to induce phenotypic changes towards more stable plaque types in atherosclerosis [22,23,24]. LDL-C lowering therapy might halt or even reverse plaque progression and improve clinical outcomes, with a clinical benefit that is proportional to the magnitude of LDL-C reduction [15,19,25].

Our results of binary logistic regression models demonstrated that patients with follow-up LDL-C < 1.4 mmol/L and a reduction ≥50% from baseline were less likely to suffer from NTL plaque progression than those failing to achieve this optimum LDL-C level. According to the current guidelines, 2019 European Society of Cardiology (ESC) and the *European Atherosclerosis Society (EAS) guideline 10 for the management of dyslipidemia*, both recommended an LDL-C target goal of <1.4 mmol/L and at least 50% reduction from baseline for secondary prevention in very high- risk ASCVD patients. Regrettably, our results showed that only 54 out of 847 (6.38%) subjects managed to reach that LDL-c recommended level during the follow-up period. There still existed a great gap between guideline and real-world practice. Our study confirms the finding of poor LDL-C control observed in the previous DYSIS and DYSIS II population [26,27].

Despite substantial reductions in LDL-C levels, 6 out of 54 (11.11%) patients with optimal follow-up LDL-C levels still developed NTL plaque progression, indicating that other factors likely took a role. A systematic and comprehensive treatment for CAD patients should not only ensure an optimal procedural result of stent implantation, but also focus on optimal pharmacological therapies such as guideline-recommended lipid-lowering treatment.

There are several limitations in the present study. First, the study was a retrospective observational study in nature although the data were prospectively collected in the clinical database. Patients who were not able to return to follow-up for 12–24 months may have been less compliant with secondary prevention post-PCI, which remains unexplored. Second, our database did not contain adequate data about the side effects of lipid-lowering drugs and the morphological changes in plaques by IVUS or OCT to fully support the beneficial effects of intensive lipid-lowering therapy. Third, the duration of 12–24 months follow-up may not be sufficient to provide information and estimates about the long-term lipid management and plaque progress in NTL in patients after PCI.

## 5. Conclusions

The present study demonstrated that achieving LDL-C < 1.4 mmol/L and a percent reduction ≥50% from baseline was associated with a significantly reduced risk of NTL plaque progression for post-PCI patients. In real-world clinical practice, more attention should be focused on the management of vascular risk factors in individuals who fail to achieve optimal level of LDL-C during follow-up periods. Our findings also provided important preliminary data to plan future randomized controlled clinical trials to formally test the hypothesis that intensive lipid-lowering therapy is effective in preventing NTL plaque progression for CAD patients receiving PCI.

## Figures and Tables

**Figure 1 jcm-12-00785-f001:**
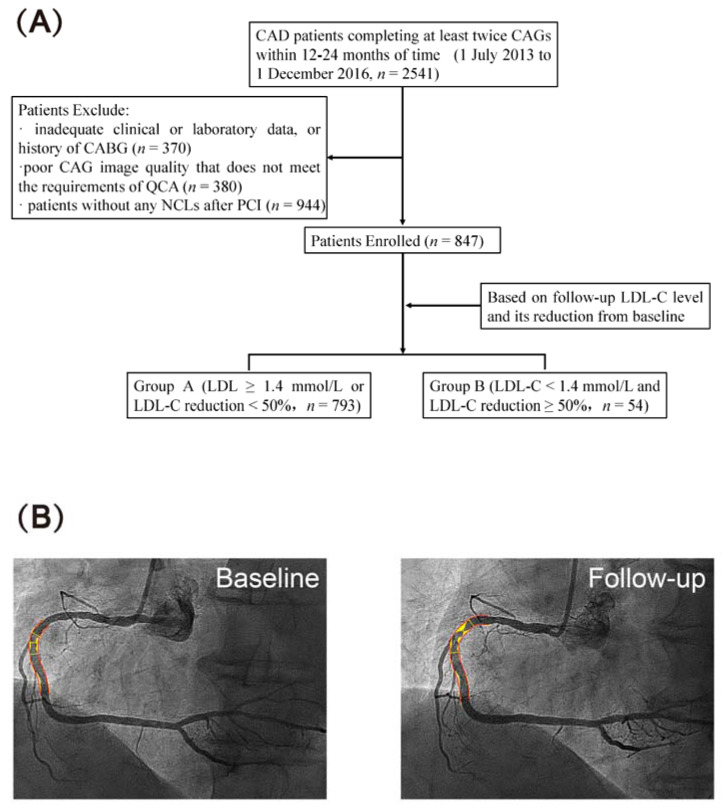
Study population and quantitative coronary angiography. (**A**) Flow chart of patient enrollment. CAG, coronary angiography; CABG, coronary artery bypass graft; QCA, quantitative coronary angiography; NTL, non-target lesion; PCI, percutaneous coronary intervention; LDL-C, low-density lipid-cholesterol. (**B**) A representative QCA result of NTL plaque progression. Angiography of the right coronary artery at Baseline and Follow-up.

**Table 1 jcm-12-00785-t001:** Baseline and follow-up characteristics of the study patients according to the follow-up LDL-C levels.

Variables	Total (*n* = 847)	Non-Optimal Group (LDL ≥ 1.4 mmol/L or LDL-C Reduction <50%, *n* = 793)	Optimal Group (LDL-C < 1.4 mmol/L and LDL-C Reduction ≥50%, *n* = 54)	*p* Value
Baseline characteristics				
Age (years)	63.13 ± 10.70	62.84 ± 11.12	65.84 ± 10.89	0.163
Male (%)	639 (75.44)	599 (75.54)	40 (74.07)	0.809
History of hypertension (%)	570 (67.30)	536 (67.59)	34 (62.96)	0.484
History of DM (%)	498 (58.80)	467 (58.89)	31 (57.41)	0.386
History of dyslipidemia (%)	158 (18.65)	150 (18.94)	8 (14.81)	0.455
History of CKD (%)	19 (2.24)	19 (2.40)	0 (0.00)	<0.001
History of ACS (%)	148 (17.47)	137 (17.28)	11 (20.37)	0.563
FPG (mmol/L)	5.64 ± 2.94	5.61 ± 2.97	6.04 ± 2.40	0.300
HbA1C (%)	4.17 ± 3.29	4.12 ± 3.29	4.79 ± 3.25	0.152
TG (mmol/L)	1.79 ± 1.36	1.79 ± 1.33	1.75 ± 1.79	0.849
TC (mmol/L)	4.08 ± 1.41	4.08 ± 1.40	4.05 ± 1.62	0.854
HDL-C (mmol/L)	1.00 ± 0.32	1.00 ± 0.32	0.98 ± 0.42	0.800
LDL-C (mmol/L)	2.47 ± 0.92	2.44 ± 0.91	2.84 ± 0.94	0.004
SCr (mmol/L)	80.38 ± 45.84	80.77 ± 46.70	74.72 ± 30.43	0.348
UA (mmol/L)	327.97 ± 106.92	328.59 ± 105.74	318.84 ± 123.72	0.517
Follow-up characteristics				
CAG Interval (days)	449.05 ± 101.54	447.74 ± 100.93	468.35 ± 109.38	0.149
FPG (mmol/L)	5.82 ± 2.70	5.81 ± 2.71	6.03 ± 2.64	0.561
HbA1C (%)	4.41 ± 3.32	4.41 ± 3.33	4.12 ± 3.21	0.920
TG (mmol/L)	1.63 ± 1.01	1.64 ± 1.00	1.52 ± 1.17	0.428
TC mmol/L)	3.60 ± 0.98	3.66 ± 0.97	2.77 ± 0.59	<0.001
HDL-C (mmol/L)	1.07 ± 0.28	1.07 ± 0.28	1.51 ± 0.28	0.683
LDL-C (mmol/L)	1.97 ± 0.72	2.03 ± 0.71	1.09 ± 0.26	<0.001
SCr (mmol/L)	83.21 ± 54.74	83.59 ± 56.33	77.64 ± 19.28	0.440
UA (mmol/L)	338.25 ± 97.43	338.75 ± 97.34	330.88 ± 99.33	0.566
NTL Plaque Progression (%)	196 (23.14)	190 (23.96)	6 (11.11)	0.007
Medications				
DAPT				
Baseline	811 (95.75)	761 (95.96)	50 (92.59)	0.361
Follow up	675 (79.69)	635 (80.08)	40 (74.07)	0.289
Statin				
Baseline	824 (97.28)	770 (97.10)	54 (100)	<0.001
Follow up	822 (97.05)	768 (96.85)	54 (100)	<0.001
β-blocker				
Baseline	604 (71.31)	565 (71.25)	39 (72.22)	0.879
Follow up	596 (70.37)	557 (70.24)	39 (72.22)	0.758
ACEI or ARB				
Baseline	497 (58.68)	462 (58.26)	35 (64.81)	0.338
Follow up	456 (53.84)	425 (53.59)	31 (57.41)	0.587

Data are expressed as mean ± SD, or frequency counts (percentages), as appropriate. DM, diabetes mellitus; CKD, chronic kidney disease; ACS, acute coronary syndrome; FPG, fasting plasma glucose; HbA1C, glycosylated hemoglobin; TG, triglycerides; TC, total cholesterol; HDL-C, high-density lipoprotein-cholesterol; LDL-C, low-density lipoprotein-cholesterol; SCr, serum creatinine; UA, uremic acid; CAG, coronary angiography; NTL, non-target lesion; DAPT, dual anti-platelet therapy; ACEI, angiotensin-converting enzyme inhibitor; ARB, angiotensin receptor blocker.

**Table 2 jcm-12-00785-t002:** QCA analysis of NTLs according to the follow-up LDL-C levels.

Variables	Non-Optimal Group (LDL ≥ 1.4 mmmol/L or LDL-C Reduction < 50%, *n* = 793)	Optimal Group (LDL-C < 1.4 mmmol/L and LDL-C Reduction ≥ 50%, *n* = 54)	*p* Value
Lesion length (mm)	16.66 ± 10.88	15.8 ± 9.85	0.462
MLD-BL (mm)	1.74 ± 0.63	1.71 ± 0.60	0.671
RVD-BL (mm)	3.00 ± 0.76	2.94 ± 0.64	0.465
DS-BL (%)	42.28 ± 13.65	42.66 ± 12.85	0.795
MLD-FU (mm)	1.56 ± 0.63	1.62 ± 0.60	0.392
DS-FU (%)	48.43 ± 15.04	46.12 ± 13.55	0.149
△DS (%)	6.15 ± 15.01	3.46 ± 12.77	0.092
Plaque Progression (%)	190 (23.96)	6 (11.11)	0.007

Data are expressed as mean ± SD, or frequency counts (percentages), as appropriate. NTL, non-target lesion; BL, baseline; FU, follow-up; MLD, minimal luminal diameter; RVD, reference vessel diameter; DS, diameter stenosis; △DS, DS-FU minus DS-BL.

**Table 3 jcm-12-00785-t003:** Logistic regression analysis for the presence of NTL plaque progression.

	Univariate		Multivariate	
Variables	OR (95% CI)	*p* Value	OR (95% CI)	*p* Value
Age	0.991 (0.977–1.005)	0.226		
Male	1.531 (1.028–2.279)	0.036	1.529 (1.026–2.279)	0.037
History of Smoking	0.987 (0.717–1.360)	0.938		
History of hypertension	1.169 (0.827–1.652)	0.376		
History of DM	1.244 (0.895–1.729)	0.194		
History of dyslipidemia	0.933 (0.616–1.413)	0.744		
History of CKD	0.883 (0.290–2.693)	0.827		
History of ACS	1.345 (0.900–2.010)	0.148		
CAG interval	1.001 (0.999–1.002)	0.223		
FU-FPG	1.036 (0.978–1.097)	0.234		
FU-HbA1C	1.032 (0.983–1.083)	0.203		
FU-TG	1.065 (0.915–1.240)	0.417		
FU-TC	1.059 (0.902–1.243)	0.483		
FU-HDL-C	0.638 (0.353–1.154)	0.137		
FU-LDL-C	1.028 (0.825–1.280)	0.808		
LDL-C < 1.4 mmol/L and LDL-C reduction ≥50%	0.397 (0.167–0.941)	0.036	0.398 (0.167–0.945)	0.037
FU-SCr	1.001 (0.998–1.003)	0.533		
FU-UA	1.000 (0.998–1.002)	0.957		

NTL, non-target lesion; DM, diabetes mellitus; CKD, chronic kidney disease; ACS, acute coronary syndrome; CAG, coronary angiography; FU, follow-up; FPG, fasting plasma glucose; HbA1C, glycosylated hemoglobin; TG, triglycerides; TC, total cholesterol; HDL-C, high-density lipoprotein-cholesterol; LDL-C, low-density lipoprotein-cholesterol; SCr, serum creatinine; UA, uremic acid.

## Data Availability

The data that support the findings of this study are available from the corresponding author on reasonable request.

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
