# Peer review of "Influence of LDL-Cholesterol Lowering on Coronary Plaque Progression of Non-Target Lesions in Patients Undergoing Percutaneous Coronary Intervention: Findings from a Retrospective Study"

_jcm, 2023, doi:10.3390/jcm12030785_

Round 1

Reviewer 1 Report

In their interesting paper, the authors describe the difference in non-culprit plaque progression in patients achieving the dual ESC/EAS LDL-C goals (optimal group) vs those who are not achieving both goals (non optimal group) following a PCI in a large multi-center real world observational cohort of Chinese patients.  The results clearly support the dual goals established in the 2019 ESC/EAS guideline, showing a clear benefit of achieving them in terms of reduced progression of coronary stenosis from non culprit plaqes. That is a great piece of RWE supporting those recommendations. However, despite achieving the dual goals, there was still a significant number of patients with plaque progression, what shows the need for additional therapy to address this residual risk. Moreover, the authors have not addressed the issue of plaque vulnerability (see below).

Major comments

1)      Have you compared the double non-optimal group with the optimal group and with the single-optimal group? Also, were there any differences between the two single non-optimal groups (one with LDL-C >= 55 mg/dL and the one with < 50% reduction?). Please include a breakdown data in a table format in the paper or in a supplement. It is particularly interesting to see what is more beneficial - a >= 50% LDL-C reduction or an LDL-C level < 55 mg/dL. I expect that a combination fo both will have an additive effect.

1)      The more prevalent plaque hypothesis accepts that subsequent events are a consequence of non-obstructive lesions with vulnerable plaques aka TCFA characteristics (JAMA Cardiol. doi:10.1001/jamacardio.2022.3926 ) and not of progression on lumen narrowing lesions (ischemia hypothesis). This needs to be discussed. Ideally, I would like to see OCT characterization on the non culprit lesions, but I understand that the technique may be not accessible to authors.  

Reviewer 2 Report

The work appears linear. The methods of analysis are adequate. The results agree with the premises. Laboratory parameters are indicated in a confusing way sometimes in mmol/l sometimes in mg/dlIt would be useful to divide the subjects treated with high efficacy versus moderate/ low efficacy statins and evaluate if there is a difference in the primary end point of the study. The two groups do not balance well 793 vs 53. This can detract from the value of the result. But it reflects the general condition of the undertreatment of these patients.  The data was collected before the release of the 2019 EAS/ESC guidelines, where the target was set at 55 mg/dl of LDL. It would be useful to do an analysis, the same with subjects with LDL < 70 mg/dl
